# Diagnostic Value of Six Thyroid Imaging Reporting and Data Systems (TIRADS) in Cytologically Equivocal Thyroid Nodules

**DOI:** 10.3390/jcm9072281

**Published:** 2020-07-17

**Authors:** Dorota Słowińska-Klencka, Kamila Wysocka-Konieczna, Mariusz Klencki, Bożena Popowicz

**Affiliations:** Department of Morphometry of Endocrine Glands, Medical University of Lodz, Pomorska Str 251, 92-213 Łódź, Poland; kamila.wysocka91@wp.pl (K.W.-K.); marklen@tyreo.umed.lodz.pl (M.K.); bozena.popowicz@umed.lodz.pl (B.P.)

**Keywords:** TIRADS, FNA, thyroid ultrasonography, thyroid cancer

## Abstract

The aim was to compare the usefulness of selected thyroid sonographic risk-stratification systems in the diagnostics of nodules with indeterminate/suspicious cytology or unequivocal cytology in a population with a history of iodine deficiency. The diagnostic efficacy of ACR-TIRADS (the American College of Radiology Thyroid Imaging Reporting and Data Systems), EU-TIRADS (European Thyroid Association TIRADS), Korean-TIRADS, Kwak-TIRADS, AACE/ACE-AME-guidelines (American Association of Clinical Endocrinologists/ American College of Endocrinology-Associazione Medici Endocrinologi guidelines) and ATA-guidelines (American Thyroid Association guidelines) was evaluated in 1000 nodules with determined histopathological diagnosis: 329 FLUS/AUS (10.6% cancers), 167 SFN/SHT (11.6% cancers), 44 SM (77.3% cancers), 298 BL (benign lesions), 162 MN (malignant neoplasms). The percentage of PTC (papillary thyroid carcinoma) among cancers was higher in Bethesda MN (86.4%) and SM (suspicion of malignancy) nodules (91.2%) than in FLUS/AUS (57.1%, *p* < 0.005) and SFN/SHT (suspicion of follicular neoplasm/ suspicion of Hürthle cell tumor) nodules (36.8%, *p* < 0.001). TIRADS efficacy was higher for MN (AUC: 0.827–0.874) and SM nodules (AUC: 0.775–0.851) than for FLUS/AUS (AUC: 0.655–0.701) or SFN/SHT nodules (AUC: 0.593–0.621). FLUS/AUS (follicular lesion of undetermined significance/ atypia of undetermined significance) nodules assigned to a high risk TIRADS category had malignancy risk of 25%. In the SFN/SHT subgroup none TIRADS category changed nodule’s malignancy risk. EU-TIRADS and AACE/ACE-AME-guidelines would allow diagnosing the highest number of PTC, FTC (follicular thyroid carcinoma), HTC (Hürthle cell carcinoma), MTC (medullary thyroid carcinoma). The highest OR value was for Kwak-TIRADS (12.6) and Korean-TIRADS (12.0). Conclusions: TIRADS efficacy depends on the incidence of PTC among cancers. All evaluated TIRADS facilitate the selection of FLUS/AUS nodules for the surgical treatment but these systems are not efficient in the management of SFN/SHT nodules.

## 1. Introduction

Preoperative diagnostics of thyroid nodules is still a subject of intensive research. One of the explored fields is the usefulness of Thyroid Imaging Reporting and Data Systems (TIRADS) in the selection of nodules for fine-needle aspiration biopsy (FNA) as well as the stratification of the risk of malignancy (RoM) related to cytologically equivocal thyroid nodules. TIRADS are based on the analysis of ultrasound malignancy risk features (US malignancy features). Separately, these features do not have both satisfying sensitivity (SEN) and specificity (SPC) at the same time, so the use of various sets of the features has been proposed. There is however some disagreement on both the significance of particular features and their optimal association. Even the term TIRADS is not used to denote each proposed ultrasound risk stratification system. But for the sake of simplicity we used that term universally in the paper. First TIRADS, created by Horvath et al. (2009) and by Park et al. (2009) were complicated and difficult to implement in the routine clinical practice. Subsequent systems were developed concurrently in Asia, Europe and the United States [1,2].

In Korea Kwak et al. (2011) proposed a much simplified TIRADS based on the assessment of a number of US malignancy features (hypoechogenicity, microlobulated or irregular margins, microcalcifications, taller-than-wide shape and solid echostructure of nodule) [3]. Each US malignancy feature was assigned the same weight despite carrying a different probability of malignancy. That system was developed into the next TIRADS of the Korean team, in which each US malignancy feature (hypoechogenicity, marked hypoechogenicity, non-parallel, microlobulated or spiculated margin, ill-defined margin and microcalcification) received a different risk score according to their odds ratio for predicting malignancy [4]. Eventually, Korean Society of Thyroid Radiology (KSThR) recommended K-TIRADS, a system in which the evaluation of a nodule’s structure (solid hypoechoic vs. other) is the first step necessary for further analysis of 3 US malignancy features of high specificity (microcalcifications, nonparallel orientation and spiculated/microlobulated margin) [5]. 

In Europe, the system based on the proposition of Horvath et al., 2009 evolved [1]. The French Society of Endocrinology published French-TIRADS, in which Horvath’s TIRADS had been simplified from the system of numerous subcategories to a system of a five-point scale [6]. Currently the European Thyroid Association (ETA) recommends a similar 5-grade system, EU-TIRADS, in which the high specificity US malignancy features include marked hypoechogenicity, irregular shape, irregular margins, microcalcifications. 

In the USA, several systems of US malignancy feature evaluation had been developed concurrently [7]. The American Thyroid Association (ATA) recommends a five-category system based mainly on the presence of high specificity US malignancy features (irregular margins, microcalcifications, taller than wide shape, rim calcifications with small extrusive soft tissue component and evidence of extrathyroidal extension) in a hypoechoic nodule [8]. The American Association of Clinical Endocrinologists (AACE), American College of Endocrinology (ACE) and Associazione Medici Endocrinologi (AME) jointly recommend a 3-grade scale. According to this system a nodule is categorized into the high-risk class when any high specificity US malignancy feature is present (marked hypoechogenicity, spiculated or lobulated margins, microcalcifications, taller-than-wide shape and presence of extrathyroidal growth or pathologic lymphadenopathy) [9]. The presence of other US malignancy features (intranodular vascularization, macro- or rim calcifications and hyperechoic spots of uncertain significance) qualifies a nodule into the intermediate-risk class. On the other hand, the American College of Radiology (ACR) assigns from 1 to 3 points to each of the analyzed US malignancy features. A nodule is classified into one of five final categories of ACR-TIRADS based on the total number of points [10]. 

Each of the abovementioned systems relates the category of ultrasound risk to nodule size thresholds for biopsy. Nodules subjected to FNA are then classified into one of six categories of cytological diagnosis according to the Bethesda System for Reporting Thyroid Cytopathology (BSRTC) [11,12]. The classification distinguishes non-diagnostic (ND) outcomes, two categories of unequivocal results: category II—benign lesions (BL) and category VI—malignant neoplasms (MN) and 3 other categories of ambiguous results. These equivocal results include the category III—follicular lesion of undetermined significance (FLUS)/atypia of undetermined significance (AUS), IV—suspicion of follicular neoplasm (SFN), suspicion of Hürthle cell tumor (SHT) and V—suspicion of malignancy (SM). In the case of an equivocal category, the clinical management depends on the nodule’s RoM as determined by the joint clinical, cytological, ultrasound and in some centers also molecular assessment. The epidemiological status of the examined population should also be considered, particularly in regard to iodine supply. Iodine deficiency modifies the relative frequency of non-neoplastic and neoplastic thyroid lesions, as well as the relative incidence of papillary thyroid carcinoma (PTC) and follicular thyroid carcinoma (FTC) [13]. These factors can influence the assessment of US malignancy feature usefulness because the ultrasound image of follicular neoplasm differs from the image of PTC [14]. Data on the efficiency of various TIRADS in populations with iodine deficiency are scarce, especially with cytological outcomes verified against histopathological examination.

The aim of the present study was to compare the usefulness of selected thyroid sonographic risk-stratification systems in the diagnostics of nodules with indeterminate/suspicious cytology or unequivocal cytology in a population with history of exposure to iodine deficiency.

## 2. Material and Methods

### 2.1. Examined Patients

FNA and ultrasound imaging examinations were performed in a single center, in the years 2010–2019, in patients referred by endocrinologists from outpatient clinics. The majority of the examined patients had been exposed to moderate iodine deficiency for most of their lives. In the 1990s our country was classified as a moderately iodine-deficient area according to the criteria of the International Council for Control of Iodine Deficiency Disorders. The mandatory iodization of household salt was introduced in 1997. The efficacy of that prophylaxis in lowering the prevalence of goiter among school-aged children below <5% was confirmed as early as 2005 [15]. Almost 90% of our patients had been exposed to moderate iodine deficiency for at least half of their life (the period of sufficient iodine supply in that group was 22 years maximum and patients under 44 constituted 10.4% of the examined group). 

The study included 1000 nodules (revealed in 866 patients) with full ultrasound imaging data, a diagnostic FNA outcome and the known result of the postoperative histopathological examination (see Appendix A). Patients previously treated surgically or with radioiodine, as well as patients with positive neck irradiation history, were excluded from the analysis. The study included all nodules classified into the categories III-VI of the Bethesda system and a set of subsequently biopsied nodules with FNA result of category II to get the total number of analyzed nodules equaled to 1000. Given that, the analysis was performed for 540 equivocal (EC) nodules, including 329 FLUS/AUS, 167 SFN/SHT, 44 SM and for 460 unequivocal (UC) nodules, including 298 BL and 162 MN (Table 1).

### 2.2. Microscopic Examination

The biopsy was performed on thyroid nodules with a diameter of at least 5 mm (and usually over 1 cm) and at least one malignancy risk factor (ultrasonographic or clinical). In most cases two aspirations of a nodule were performed. Smears were fixed with 95% ethanol solution and stained with haematoxylin and eosin. Surgical thyroidectomy specimens were processed using standard procedures. 

A detailed description of the classification of nodules into specific diagnostic categories of the Bethesda system was presented in our earlier report [16]. In particular, the category IV did not included lesions with nuclear features of PTC. In the case where the specimen showed features of both the category II and IV, the category III was assigned. Biopsies with a presence of local features suggestive of PTC (nuclear grooves, enlarged nuclei with pale chromatin and alterations in nuclear contour and shape) in an aspirate that was otherwise benign in microscopic appearance or specimens with limited cellularity but with nuclear atypia were rarely classified into the category III of the BSRTC. Patients with a cytological outcome of SFN/SHT, SM or MN were routinely referred for the surgical treatment. In the case of a diagnosis of BL or FLUS/AUS, the surgical treatment was performed based on the patient’s preference or due to the large size of the goiter as well as the presence of other clinical risk features. The histopathologic examination was performed according to the standard procedure and its results were formulated according to the WHO classification of thyroid tumors that was in effect at the time of examination. The reclassification of the histopathological examination in order to reveal cases of non-invasive follicular thyroid neoplasm with papillary-like nuclear features (NIFTP) was not performed. The only case of NIFTP diagnosed after the introduction of this category was excluded from the analysis. 

The histopathological examination confirmed all unequivocal FNA results (BL and MN) and revealed cancers in 10.6% of FLUS/AUS nodules, 11.6% of SFN/SHT nodules and 77.3% of SM nodules. The percentage of PTC among cancers was significantly higher in cytologically MN (86.4%) and SM nodules (91.2%) than in FLUS/AUS nodules (57.1%, *p* < 0.005) and SFN/SHT nodules (36.8%, *p* < 0.001). 

### 2.3. Analysis of US Malignancy Features

The analysis of US malignancy features was done prospectively. The presence of particular US malignancy features was assessed by experienced sonographers (three doctors with over 20 years of experience who performed 90% of reports and two others with ten years’ experience) directly before FNA, according to a unified pattern that’s been used at our department for many years. We used a computer system dedicated for collecting detailed information on examined nodules in a database. The system had been created by one of the authors of the study—MK. On the basis of these data, three diameters of biopsied nodules were determined as well as the presence of: (1) marked hypoechogenicity (compared to the echogenicity of the strap muscles); (2) hypoechogenicity (as compared to the normal thyroid); (3) solid echostructure (>90% solid) (4) more solid than cystic echostructure (>50% solid); (5) suspicious shape/orientation—taller than wide; (6) suspicious margins—irregular (including microlobulated, spiculated and suggesting extrathyroidal extension); (7) microcalcifications; (8) macrocalcifications without microcalcifications; (9) isolated rim calcifications (each type of calcification was assessed separately and there was a possibility to evaluate their combinations); (10) pathological vascularization (marked intranodular vascular spots). The presence of other ultrasound features was assessed as follows: (1) more (or equally) cystic than solid echostructure including purely cystic echostructure and mostly cystic structure with reverberation artifacts; (2) spongiform echostructure (>50% of nodule, without obvious solid areas). The US examinations were performed with the use of the Aloka Prosound Alpha 7 ultrasound system, ALOKA co. Ltd., Tokyo, Japan with a 7.5–14 MHz linear transducer.

With the use of the set of features specified above, all thyroid nodules were classified into specific categories of six risk stratification systems: EU-TIRADS (EU-T) [7]; K-TIRADS (K-T) [5]; ACR-TIRADS (ACR-T) [10], the system developed by Kwak (Kw-T) [3] and the systems recommended by ATA (ATA-T) [8] and by ACCE, ACE and AME (3A-T) [9]. Two researchers (KWK and DSK) independently assigned all the ultrasound features for TIRADS score calculation. In the case of discrepancy (which occurred in 39 nodules), the US report was jointly reevaluated and discussed to confirm its categorization. In the case of the ATA-T system, a modification has been applied, because this system does not cover all ultrasound nodules patterns; in particular it lacks patterns in which iso- or hyperechoic nodules show high malignancy risk features. In total, 51 (5.1%) nodules did not satisfy the criteria of ATA-T classification and those nodules corresponded to 16 (31.4%) cancers (14 PTC and 2 FTC) and 35 (68.6%) benign lesions. The FNA results of those nodules were classified into the category II, 14 cases; category III, 16 cases (including 2 PTC); category IV, 8 nodules (including 1 PTC); category V, 1 nodule (FTC); and category VI, 12 nodules (11 PTC, 1 FTC). We decided to classify such nodules into the highly suspicious category. That allowed us to compare how all systems work in an evaluation of the same set of nodules. We also assessed the diagnostic efficacy of ATA-T with the nodules in question excluded. We had not identified disrupted rim calcifications with small extrusive soft tissue component as a separate feature (which is included in ATA-T), but the nodules presenting such an image were treated as ones with irregular margins what resulted in the same output of the categorization. 

### 2.4. Analyses, Statistical Evaluation

The incidence of US malignancy features was assessed in the nodules classified into particular diagnostic categories of FNA in respect of the division of the nodules into benign lesions and cancers in the postoperative histopathological examination. The associations between individual US malignancy features and malignancy were evaluated with the use of logistic regression analysis in a group of nodules with unequivocal cytology (UC—categories II and VI) and equivocal cytology (EC—categories III-V). Odds ratios (OR) with relative 95% confidence intervals (95% CI) were calculated to determine the relevance of all potential predictors of the outcome.

Next, the distribution of benign and malignant nodules among particular categories of the examined TIRADS was assessed. That lead to the determination of RoM for nodules in each of the TIRADS categories—T-RoM (the proportion of cancers among all nodules in each category)—in the entire examined set of nodules and in relation to the FNA result. We calculated how T-RoM of a nodule influenced its RoM related to the class of FNA outcome (FNA-RoM). For each examined TIRADS, separately in the UC group and in the subgroups of the EC group, the categories that showed the highest efficiency in the classification of benign and malign lesions were identified by the analysis of the receiver operating characteristics curve (ROC) and the area under the ROC (AUC) value. The effectiveness of the determined thresholds in all groups was presented as SEN, SPC, the accuracy (ACC), the negative predictive value (PPV) and the negative predictive value (NPV). The percentage of nodules that satisfied the given criteria was also determined. The odds ratio for the established cut-off values was assessed with the use of logistic regression analysis.

The statistical analysis was performed with Statistica, version 10 statistical software. The comparison of frequency distributions was performed with χ^2^ test (with modifications appropriate for the number of analyzed cases). The Kruskal–Wallis test was used for comparing continuous variables between groups. The value of 0.05 was assumed as the level of significance. The study design was approved by the Local Bioethics Committee and all patients gave their informed consent.

## 3. Results

The incidence of individual US malignancy features in UC nodules and in particular subgroups of EC nodules in relation to the histopathological outcome: thyroid cancer vs. benign lesion has been shown in Appendix A. In the UC group all US malignancy features occurred more often in cancers than in benign nodules with the exception of macrocalcifications (without micro-) and isolated rim calcifications. The logistic regression analysis confirmed that the presence of any of 7 examined US malignancy features was an independent feature in the differentiation between benign and malignant UC nodules: marked hypoechogenicity (OR: 9.8, CI95%: 3.7–26.1, *p* < 0.0001), hypoechogenicity (OR: 4.0, CI95%: 2.0–8.0, *p* < 0.0001), solid echostructure (OR: 3.3, CI95%: 1.2–8.9, *p* < 0.05), suspicious shape (OR: 4.0, CI95%: 1.6–9.8, *p* < 0.005), suspicious margins (OR: 6.8, CI95%: 3.0–15.5, *p* < 0.0001), microcalcifications (OR: 14.9, CI95%: 4.5–49.7, *p* < 0.0001) and pathological vascularization (OR: 2.3, CI95%: 1.1–4.9, *p* < 0.05). In the EC group, only the marked hypoechogenicity differed significantly between cancers and benign nodules in all the subgroups. The suspicious margins occurred more often in cancers than in benign lesions in FLUS/AUS and SM nodules, while the same was true for microcalcifications only in the FLUS/AUS subgroup. Microcalcifications and suspicious margins were independent risk factors in the FLUS/AUS subgroup (OR: 6.9, CI95%: 2.2–21.6, *p* < 0.005 and OR: 3.7, CI95%: 1.1–11.8, *p* < 0.05, respectively) while the marked hypoechogenicity was such a factor in the SM subgroup (OR: 4.4, CI95%: 1.4–13.4, *p* < 0.01).

Table 2 shows the distribution of benign and malignant nodules between particular categories of analyzed systems, T-RoM of these categories and AUC value characterizing each system. T-RoM related to particular categories was concordant with expectations with the exception of the high suspicion category of ATA-T, in which case it was lower than expected (without our modification including iso- or hyperechoic nodules with high malignancy risk features T-RoM increased to 62.3%) as well as low risk category EU-T, low suspicion for malignancy category of Kw-T and mildly suspicious category of ACR-T where T-RoM was higher than expected. 

The diagnostic efficacy of the evaluated systems, as measured by AUC, was in the range from 0.763 for 3A-T up to 0.793 for Kw-T for the whole examined set of nodules. The efficacy was higher in the groups with high percentage of PTC among cancers (UC and SM) than in FLUS/AUS or SFN/SHT group. For the latter, none of the evaluated systems showed significant efficacy as measured by AUC (Table 3). The exclusion of nodules other than hypoechoic from the category 5 of ATA-T system decreased its AUC in FLUS/AUS and SFN/SHT nodules and increased it in UC and SM nodules.

In the UC group, the assignment of a nodule to the highest malignancy risk category in each TIRADS was related to the RoM of the nodule significantly higher than its initial FNA-RoM. (In the case of Kw-T, system that increased RoM occurred even at the level of the category 4c). In the EC group, the assignment of a nodule to the mentioned categories increased the RoM of the nodule but significant differences were observed for all TIRADS only in FLUS/AUS nodules. In the SFN subgroup, none of TIRADS significantly increased nodule’s RoM, and the closest to the border of significance was the 11.2% increase for Kw-T system with the threshold at the category 4c. In the SM group, nodule’s RoM increased to 100% for all evaluated systems, but the difference between T-RoM and FNA-RoM was significant only for ATA-T and EU-T systems. In the UC group, the assignment of a nodule to any of 2 categories of the lowest malignancy risk in 3A-T, K-T and Kw-T systems or any of 3 such categories in ATA-T, ACR-T and EU-T systems significantly lowered the RoM of the nodule. In any of the subgroups of the EC group, the similar effect was not observed irrelatively of the threshold level applied. 

Table 4 shows the data on the diagnostic efficiency TIRADS systems (for the categories which proved to serve as thresholds characterized by the highest values of ACC). The highest SEN in all the groups was noted for EU-T and 3A-T systems (UC: 77.8%, SM: 61.8%, FLUS/AUS: 51.4%, SFN/SHT: 52.6%), and the lowest one for ACR-T. In the UC group as well as SM and FLUS/AUS subgroups all the systems showed over 80% SPC, and SPC over 90% was found for K-T, Kw-T and ACR-T. In the group of SFN/SHT nodules, only K-T, Kw-T and ACR-T showed over 80% SPC. In all the groups of nodules, the highest sum of SEN and SPC was noted for EU-T and 3A-T systems. 

With the threshold levels lowered one grade (set at the category 4 of K-T, EU-T, ATA-T, ACR-T systems, category 2 of 3A-T, category 4b of Kw-T; see Appendix A) in the UC group, all the systems had over 90% SEN (the highest, 3A-T: 100.0% and EU-T: 96.3%). SPC was within the range of 54.4–62.8% for all the evaluated systems with the exception of 3A-T, for which it amounted to 16.4%. In subgroups of the EC group all the systems showed at least 80% SEN at those same thresholds. SPC was within the range of 29.9–41.8% for FLUS/AUS nodules, 20.3–24.3% for SFN/SHT nodules and 50.0% for SM nodules with the exception of 3A-T. The 3A-T system had a very low SPC at those thresholds (SM nodules: 10.0%, FLUS/AUS nodules: 3.7%, SFN/SHT nodules: 0.7%).

Table 5 shows the number of revealed cancers and their types when the thresholds levels for performing FNA are set to obtain the maximum possible ACC in each evaluated system. The 3A-T and EU-T systems would allow to diagnose the highest number of all cancers and the highest number of cancers of each type (PTC, FTC, HTC and MTC), but the percentage of biopsied nodules would also be the highest (32.2%). The ACR-T system would permit to decrease the number of performed FNA by 13.6% but SEN would be lower by 22.0% when compared with 3A-T and EU-T. The highest increase in RoM of a nodule at the indicated thresholds was found for Kw-T and K-T systems (OR was 12.6 and 12.0, respectively). The OR value would be the highest for those systems even when 51 nodules not satisfying ATA-T criteria had been excluded from the analysis. 

SEN of all evaluated systems was higher for PTC than for FTC or HTC. In all the systems the majority of PTC was classified into the highest risk category or 4c Kw-T category, while the majority of FTC and HTC were located in the categories of one grade lower risk (Figure 1).

## 4. Discussion

The comparison of data on the efficacy of various TIRADS systems and the efficacy of a single TIRADS among different populations is not simple. Kim et al. (2019) performed a meta-analysis including 4 systems: ACR-T, ATA-T, K-T and EU-T and concluded that the overall diagnostic performance of the four US-based risk stratification systems was comparable, but also that pooled SEN and SPC were the highest for EU-T [17]. On the other hand, Castellana et al. showed in their meta-analysis that SEN of EU-T in the selection of nodules for FNA was markedly lower than for the 4 other analyzed systems (3A-T, ATA-T, K-T and ACR-T) [18]. There are large differences between particular reports concerning the threshold levels for various systems, the methods used for verification of final diagnoses (histopathology vs. cytology vs. clinical follow up) as well as the selection of nodules for the analysis in relation to the category of FNA result. In the majority of papers, including our study, non-diagnostic FNA were excluded from the analysis, however in a number of studies the nodules with indeterminate and suspicious cytology were also excluded [19]. The latter have an enormous impact on obtained results, because indeterminate categories of the Bethesda classification usually include nodules corresponding to FTC and HTC. Our studies, as well as other researchers’ reports, showed that ultrasound image of these cancers differs from the image of PTC [14,18,20,21,22]. It is important to emphasize that difference because US malignancy features and TIRADS systems have been established mainly on the basis of the ultrasound image of the most common PTCs [3,4,21]. Unsurprisingly, we showed that efficacy of US malignancy features, and consequently the efficacy of TIRADS systems lowers following the decrease in the percentage of PTC among cancers. The evaluated TIRADS had the comparable and good efficacy in cytological groups of MN and SM nodules (containing 86.4% and 91.2% PTC among cancers, respectively) and the markedly lower effectiveness in the groups of nodules with indeterminate cytology. It was the lowest in SFN/SHT nodules. The percentage of PTC among cancers in that group did not exceed 40%, and RoM of those nodules was under 15%. Such low values are the product of two factors. The first one is the epidemiological situation of our population, which has been exposed to iodine deficiency for many years. For that reason, SFN/SHT nodules correspond mainly to non-neoplastic follicular lesions developing in a consequence of iodine shortage, and the ratio of PTC to FTC is lower [23]. The other factor in question is a conservative attitude to the classification of smears into the category IV of the Bethesda system. The pathologists at our center avoided assigning smears with cells presenting features of PTC into the class IV, in spite of the fact that since 2017 follicular-patterned cases with mild nuclear changes can also be classified into this class of diagnoses [12]. With such a low percentage of PTC in the group of SFN/SHT nodules, the evaluated TIRADS proved to be inefficient in distinguishing between benign and malignant lesions. Categorization of a SNF/SHT nodule into the highest risk category in any TIRADS did not significantly increase its RoM. Similarly, categorization of a nodule to the lowest risk category of TIRADS did not significantly decreased RoM of an SFN/SHT nodule (but it is worth noting that none of the examined cancers was classified into such a category).

In the case of the category III, the diagnostic efficacy of US malignancy features and TIRADS was higher. The FNA-ROM of FLUS/AUS and SFN/SHT nodules was similar, but the percentage of PTC among cancers was higher in FLUS/AUS nodules than in SFN/SHT nodules by over 20 percentage points. Consequently, the classification of a nodule into the category 4c or higher of Kw-T system or a high-risk category in other TIRADS resulted in a significant increase of RoM of the nodule up to the level that justified the surgical treatment. This observation is of particular importance in light of the fact that in many patients with a category III nodule, the result of the repeated FNA does not help in finding optimal clinical management. 

Reports on the usefulness of TIRADS in making clinical decisions in patients with indeterminate cytology are not easily comparable. There are differences in the selection of nodules for the analysis and in their FNA-RoM, particularly in relation to FLUS/AUS nodules as their FNA-RoM can range from several up to 70 per cent [24]. In the centers where the category III is dominated by smears with nuclear atypia, including features typical of PTC, the RoM, the percentage of PTC and the efficacy of TIRADS are higher than in populations similar to ours, where the category III usually includes nodules with disturbed cellular architecture from the border between categories II and IV [25,26]. Because of the aforementioned differences, the expectations from TIRADS system and real possibilities of their use are also different. Grani et al. (2017) showed the usefulness of ATA-T and the older version of K-T for excluding malignancy in indeterminate thyroid nodules (class TIR3 of Italian Consensus for Thyroid Cytology) [27]. Tang et al. (2017) found that the ATA-T system is useful to predict malignancy in FLUS/AUS nodules [28]. Similar results were reported by Kamaya et al. from a study on Kw-T [29], while Lee et al. found ATA guidelines to be useful only in AUS subcategory of the category III [30]. Analogous conclusions were drawn be Yoon JH et al. in relation to Kw-T [26]. Hong et al. (2017) reported, as we do, that a high-suspicion US pattern in K-T system significantly increased the malignancy risk of FLUS/AUS nodules but not of SFN/SHT nodules [31]. On the other hand, Valderrabano et al. (2018) suggested that ATA sonographic patterns should be used to individualize management after the biopsy of both FLUS/AUS and SFN nodules (they did not find any differences in the distribution of histological diagnoses of malignancy between those categories) [32]. Similar conclusions were formulated by Ahmadi et al. (2019) and Barbosa et al. (2019) in relation to ATA-T and ACR-T, but the latter research group considered the category IV of the Bethesda system together with the category V (FNA-RoM in those joined categories was 61.5%) [33,34]. Yang et al., like us, did not find ACR-T, ATA-T or K-T to be useful for the RoM assessment in nodules of the category IV [35]. Chaigneau et al. (2015) found that the risk stratification with French TIRADS (which resembled the current EU-T) was significant only in Bethesda V nodules, but not in Bethesda III and IV nodules [36]. 

In the whole examined sample (nodules with unequivocal and equivocal cytology together), the diagnostic efficacy of evaluated TIRADS was comparable but the systems Kw-T, K-T and EU-T had slightly higher AUC than other systems. At the threshold levels that guaranteed the maximization of ACC (4c category of Kw-T and high-risk category of other TIRADS) EU-T and 3A-T system were characterized by the highest SEN. These systems allowed to diagnose a larger number of cancers than all other TIRADS, including cancers with diameters over 1 cm. The 3A-T and EU-T were more efficient in revealing both PTC and other common carcinomas (FTC, HTC and MTC). However, it should be noted that with the indicated threshold levels used as a criterion for the classification to perform FNA, the number of performed biopsies would be the highest for 3A-T and EU-T. On the other hand, ACR-T system would allow to limit the number of performed biopsied to the highest degree with preserved high SPC but significantly diminished SEN. Other authors also indicated similar features of that system [37,38,39,40]. The most favorable relation between the number of revealed cancers and the number of examined nodules was found for Kw-T and K-T systems, which presented the highest OR values.

In our material, the categories of the highest risk in 3A-T and EU-T systems included exactly the same nodules because the classification rules for those categories were very similar. In the case of the 3A-T, an additional criterion (apart from the presence of: marked hypoechogenicity, taller than wide shape, irregular margins and microcalcifications) was the presence of extrathyroidal growth, but we never observed features of extrathyroidal growth not accompanied by other features of high RoM. The differences between nodules classified to the highest risk categories of those systems were similarly minimal in the study by Grani et al. (2019) [37]. The advantage of EU-T system over 3A-T system consists in the number of degrees (4) in relation to the nodule’s features. It leads to a better stratification of the risk for a nodule and allows choosing the threshold at the level of category 4 or 5 depending on the aim of a user of this system—optimization of SEN vs. optimization of SPC. Timborli et al. also indicated the discrimination value of EU-T [41] and Dobruch-Sobczak et al. reported high SEN of that system in a population that had been exposed to iodine deficiency [42]. Advantages similar to that of EU-T can be also found for K-T, ATA-T, ACR-T and Kw-T. In the case of 3A-T when the threshold is set at the category 2 nearly 100% SEN is reached but this comes with unsatisfactorily low SPC (<20%). 

Comparisons of the usefulness of various TIRADS assessed in a single group of patients/nodules bring some repeating conclusions that are concordant with our observations. High SEN and AUC of K-T and Kw-T systems as well as high SPC of ACR-T along with its somewhat lower SEN than in other systems are usually underlined [37,38,39]. Xu et al. observed the highest SEN for ACR-T, but with the threshold level set at a category one step lower than the one used for other systems (EU-T and K-T) [43]. Lauria Pantano et al. found, with the use of AUC, an advantage of ACR-T system over ATA-T and 3A-T in detecting nodules with high cytological risk of malignancy, but they did not verify these diagnoses against the results of postoperative histopathologic examination [44]. In our study, none of those three systems showed significant differences in AUC, although the exclusion of nodules that did not satisfy the ATA criteria gave a significantly higher AUC value for ATA-T than for ACR-T. Similar results were reported by Gao et al. [38]. 

Another common pattern between the published results is a relation between AUC and the percentage of PTC among all cancers in the studied material. A study by Shen at al. based on a sample with a very high percentage of PTC among cancers (95.5%) showed AUC in the range of 0.869-0.896 for ACR-T, ATA-T, EU-T and Kw-T [39], while a study by Grani et al., with 75% of PTC among cancers determined AUC for five examined TIRADS within much lower values from 0.55 to 0.70 [37]. Trimboli et al. compared the usefulness of three sonographic risk-stratification systems (including ATA-T and 3A-T) in a sample of nodules with indeterminate cytology (101 lesions, 21% of malignant neoplasm, with 57% of PTC). The authors also confirmed that these systems have low ACC in such groups of nodules [45].

The values of T-ROM determined by us for particular categories of the analyzed systems generally fell within the recommended/expected ranges. The differences concerned the high-risk category of ATA-T. The risk of malignancy in our study was lower than expected and similar to that reported by Persichetti et al. for the Italian population [46] and by Rosario et al. for the Brazilian population [47]. It resulted in part from the inclusion nodules other than hypoechoic to that category. T-ROM of those additional nodules amounted to 31.4% and was close to that found for iso-/hyperechoic solid nodules with at least one suspicious sonographic feature in a study by Gao et al. (2018) (25.9%) but higher than that reported by these authors for partially cystic nodules [48]. Another reason is a notable fraction of FTC and HTC in our material—in total amounting to 11.6% of all cancers. These cancers are usually classified into lower risk categories of TIRADS. Consequently, in our material T-RoM higher than expected was determined for low-risk category of EU-T (6.7% vs. 2–4%) and mildly suspicious category of ACR-T (8.9% vs. 5%). Less prominent differences were noted for not suspicious and moderately suspicious categories of ACR-T.

There is a limitation related to the design of our study, which should be considered while interpreting its results, and it is the way of selecting nodules for analysis on the basis of the postoperative histopathological examination. However, that is also a major strength of the present study, i.e., certainty of the correct diagnosis of benign and malignant lesions. Another important advantage of our study is performing US malignancy feature evaluation directly prior to biopsy. Therefore, the result of FNA does not influence that evaluation. A further limitation of our study is the low number of cancers in the FLUS/AUS and SFN/SHT subgroups, but this number reflects the low risk of malignancy in such nodules in a population that has been exposed to iodine deficiency. 

## 5. Conclusions

The efficacy of TIRADS depends on the incidence of PTC among cancers and is generally lower for nodules with indeterminate cytology than for nodules with unequivocal cytology. All evaluated TIRADS aid in selection of FLUS/AUS nodules for the surgical treatment in a population characterized by the low risk of malignancy in nodules with indeterminate cytology and by a low percentage of PTC among cancers. However, these systems are not efficient in the management of SFN/SHT nodules in such populations. While the diagnostic efficacy of all TIRADS is similar, three of them show some shortcomings when compared to the others. The ATA guidelines do not cover all nodules. A disadvantage of 3A-T system is that no threshold level that maximizes SEN also gives acceptable SPC. A weakness of ACR-T system, at the threshold level at the category of the highest risk, is that the pressure on SPC maximization is too large while SEN remains unsatisfactory low. The greatest versatility in relation to various types of cancers is characteristic of EU-T system.

## Figures and Tables

**Figure 1 jcm-09-02281-f001:**
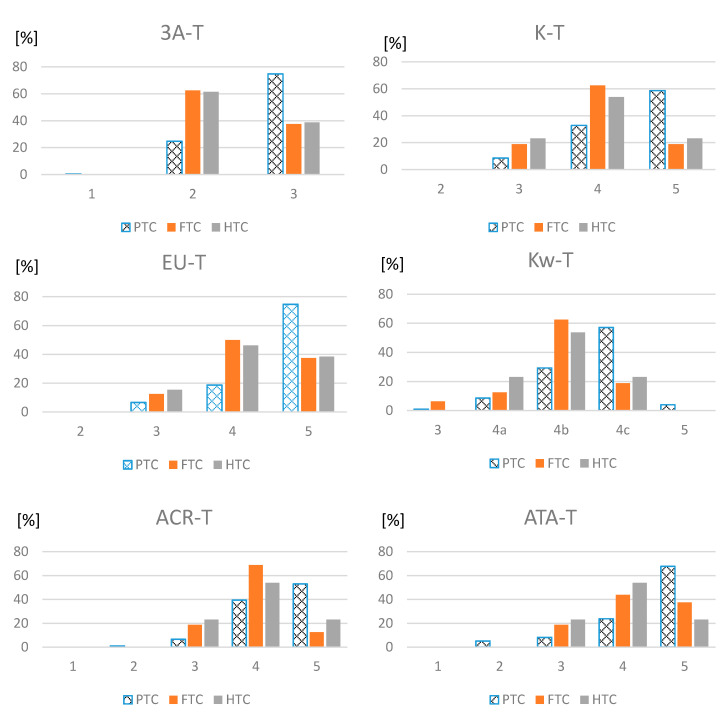
Distribution of papillary (PTC), follicular (FTC) and Hürthle cell (HTC) thyroid carcinomas among particular categories of evaluated TIRADS. 3A-T, American Association of Clinical Endocrinologists (AACE), American College of Endocrinology (ACE) and Associazione Medici Endocrinologi (AME) Thyroid Imaging Reporting and Data Systems; K-T, Korean Thyroid Imaging Reporting and Data Systems; EU-T, European Thyroid Association TIRADS; Kw-T, Kwak Thyroid Imaging Reporting and Data Systems; ACR-T, the American College of Radiology Thyroid Imaging Reporting and Data Systems); ATA-T, American Thyroid Association.

**Table 1 jcm-09-02281-t001:** Demographic data of the patients and the percentage of cancers revealed in the nodules with unequivocal (UC) and equivocal (EC) FNA results.

Parameter	Category of FNA					*p*
	UC (460)		EC (560)			
	BL	MN	FLUS/AUS	SFN/SHT	SM	
Number of nodules	298	162	329	167	44	
Number of patients	240	141	290	152	43	
Age—mean ± SD [years]	54.7 ± 11.6	50.3 ± 13.9	53.7 ± 13.6	54.1 ± 14.8	56.4 ± 14.4	*p* < 0.01 MN vs. others
No/% of males	18/7.5	20/14.2	32/11.0	15/9.9	5/11.6	NS
Volume of nodules mean ± SD [cm^3^]	7.9 ± 15.4	4.6 ± 13.9	6.6 ± 13.6	5.9 ± 12.9	3.2 ± 5.6	NS
No of Ben/Mal nodules < 1 cm	16/0	0/47	13/1	22/2	1/11	
No/% of cancers	0/0.0	162/100.0	35/10.6	19/11.4	34/77.3	*p* < 0.0001 MN & SM vs. others
No/% of PTCs among cancers	0/0.0	140/86.4	20/57.1	7/36.8	31/91.2	*p* < 0.005 MN & SM vs. FLUS/AUS, SFN/SHT
Other cancers (No/%)	-	FTC (3/1.9)	FTC (7/20.0)	FTC (5/26.3)	FTC (1/2.9)	
		HTC (1/0.6)	HTC (4/11.4)			
		PDTC (2/1.2)			HTC (1/2.9)	
		AC (1/0.6)	AC (1/2.8)	HTC (7/36.8)		
		MTC (13/9.0)	MTC (2/5.7)		MTC (1/2.9)	
		ST (2/1.2)	ANG (1/2.8)			

BL, benign lesion; FLUS/AUS, follicular lesions of undetermined significance/atypia of undetermined significance; SFN/SHT, suspicion of follicular neoplasm/suspicion of Hürthle cell tumor; SM, suspicion of malignancy; MN, malignant neoplasm; PTC, papillary thyroid carcinoma; MTC, medullary thyroid carcinoma; FTC, follicular thyroid carcinoma; HTC, Hurthle cell thyroid carcionoma; PDTC, poorly differentiated thyroid carcinoma; AC, anaplastic carcinoma; ST, secondary tumor; ANG, angiosarcoma; Ben, benign lesion in histopathological outcome; Mal, thyroid malignancy in histopathological outcome.

**Table 2 jcm-09-02281-t002:** Distribution of benign and malignant nodules between particular categories of Thyroid Imaging Reporting and Data Systems (TIRADS), the comparison of expected T-ROM with calculated T-ROM for each TIRADS and diagnostic efficacy of evaluated TIRADS as measured with AUC (TIRADS categories corresponding to the lack of nodules have been omitted).

Category of TIRADS/Guideline		Expected T-RoM	Calculated T-RoM	Mal./Ben. Nodules	AUC (95%CI)
3A-T	1—low-risk thyroid lesion	1	1.6	1/62	0.763
	2—intermediate-risk thyroid lesion	5–15	12.0	74/541	(0.728–0.798)
	3—high-risk thyroid lesion	50–90	54.3	175/147	*p* < 0.0001
K-T	2—benign	<3	0.0	0/51	0.788 ^c^ (0.755–0.821) *p* < 0.0001
	3—low suspicion	3–15	7.8	25/295	
	4—intermediate	15–50	21.5	93/340	
	5—high suspicion	>60	67.3	132/64	
EU-T	2—benign	0	0.0	0/47	0.784 ^d^ (0.752–0.816) *p* < 0.0001
	3—low risk	2–4	6.7	17/238	
	4—intermediate risk	6–17	15.4	58/318	
	5—high risk	26–87	54.3	175/147	
Kw-T	3—probably benign	0	2.53	3/116	0.793 ^a,b^ (0.760–0.825) *p* < 0.0001
	4a—low suspicion for malignancy	2–3	9.3	24/235	
	4b—intermediate suspicion for malignancy	7–38	20.5	86/333	
	4c—moderate concern, not classic for malignancy	21–92	66.7	128/64	
	5—highly suggestive of malignancy	89–98	81.8	9/2	
ACR-T	1—benign	–	0.0	0/48	0.771 (0.738–0.804) *p* < 0.0001
	2—not suspicious	<2	3.0	2/64	
	3—mildly suspicious	5	8.9	20/204	
	4—moderately suspicious	5–20	22.7	108/368	
	5—highly suspicious	>20	64.5	120/66	
ATA-T	1—benign	<1	0.0	0/1	0.778 (0.746–0.811) *p* < 0.0001
	2—very low suspicion	<3	1.2	1/81	
	3—low suspicion	5–10	8.4	24/260	
	4—intermediate suspicion	10–20	19.9	72/290	
	5—high suspicion	70–90	56.5	153/118	

a, *p* < 0.05 vs. 3A-T, ATA-T; b, *p* < 0.005 vs. ACR-T; c, *p* < 0.05 vs. 3A-T, ACR-T; d, *p* < 0.0001 vs. 3A-T.

**Table 3 jcm-09-02281-t003:** Diagnostic efficacy of evaluated TIRADS as measured with AUC in the UC group and subgroups of the EC group; the change from FNA-ROM of a nodule in relation to its TIRADS category.

TIRADS/Guideline Category		EC									UC		
		FLUS/AUS			SFN/SHT			SM			BL & MN		
		FNA-RoM: 10.6%			FNA-RoM: 11.4%			FNA-RoM: 77.3%			FNA-RoM: 35.2%		
		AUC	T-RoM	FNA-RoM vs. T-RoM	AUC	T-RoM	FNA-RoM vs. T-RoM	AUC	T-RoM	FNA-RoM vs. T-RoM	AUC	T-RoM	FNA-RoM vs. T-RoM
		*p*		*p*	*p*		*p*	*p*		*p*	*p*		*p*
3A-T	1	0.674 <0.005	0.0	NS	0.613 NS	0.0	NS	0.813 <0.0001	50.0	NS	0.827 <0.0001	0.0	<0.0001
	2		6.9	NS		8.1	NS		57.1	NS		15.2	<0.0001
	3		25.0	<0.005		18.2	NS		100.0	<0.05		72.4	<0.0001
K-T	2	0.692 <0.0001	0.0	NS	0.603 NS	0.0	NS	0.803 <0.0001	0.0	NS	0.864 ^d,e^ <0.0001	0.0	<0.0001
	3		5.2	NS		7.9	NS		60.0	NS		6.4	<0.0001
	4		10.2	NS		9.9	NS		66.7	NS		37.3	NS
	5		34.3	<0.001		22.2	NS		100.0	NS		82.8	<0.0001
EU-T	2	0.693 <0.0001	0.0	NS	0.605 NS	0.0	NS	0.851 ^f^ <0.0001	0.0	NS	0.855 ^d^ <0.0001	0.0	<0.0001
	3		4.9	NS		9.4	NS		50.0	NS		4.5	<0.0001
	4		7.9	NS		7.6	NS		64.3	NS		25.4	<0.05
	5		25.0	<0.005		18.2	NS		100.0	<0.05		72.4	<0.0001
Kw-T	1	0.681 <0.0005	3.2	NS	0.621 ^h^ NS	0.0	NS	0.790 <0.0001	50.0	NS	0.874 ^a,b,c^ <0.0001	0.0	<0.0001
	2		6.1	NS		8.8	NS		57.1	NS		9.2	<0.0001
	3		9.8	NS		9.1	NS		66.7	NS		35.9	NS
	4		32.4	<0.001		25.0	NS		100.0	NS		82.3	<0.0001
	5		50.0	NS		0.0	NS		100.0	NS		100.0	<0.005
ACR-T	1	0.655 ^g^ <0.005	0.0	NS	0.593 NS	0.0	NS	0.775 <0.0005	0.0	NS	0.857 ^d^ <0.0001	0.0	<0.0001
	2		0.0	NS		0.0	NS		66.7	NS		0.0	<0.0001
	3		6.8	NS		10.0	NS		50.0	NS		8.0	<0.0001
	4		11.0	NS		9.6	NS		76.2	NS		36.5	NS
	5		27.3	<0.05		21.4	NS		100.0	NS		82.1	<0.0001
ATA-T	1	0.701 <0.005 0.652 *	-	-	0.589 NS 0.554 *	-	-	0.810 <0.0001 0.847 *	-	-	0.843 <0.0001 0.862 *	0.0	NS
	2		0.0	NS		0.0	NS		50.0	NS		0.0	<0.0001
	3		5.6	NS		8.1	NS		55.6	NS		7.6	<0.0001
	4		8.9	NS		10.2	NS		61.5	NS		36.5	NS
	5		28.1	<0.001		17.5	NS		100.0	NS		71.4	<0.0001

*, Value of AUC after the exclusion of non-hypoechoic nodules from the ATA-T category 5; a, *p* < 0.001 vs. ATA-T, 3A-T; b, *p* < 0.005 vs. K-T; c, *p* < 0.05 vs. ACR-T; d, *p* < 0.005 vs. 3A-T; e, *p* < 0.05 vs. ATA-T; f, *p* < 0.05 vs. ATA-T, ACR-T, K-T, Kw-T; g, *p* < 0.05 vs. K-T, ATA-T; h, *p* < 0.005 vs. ATA-T.

**Table 4 jcm-09-02281-t004:** Data on the diagnostic efficacy of analyzed TIRADSs in examined groups of nodules—data for the thresholds that gave the highest ACC values.

TIRADS/Guideline Threshold Category		SEN	SPC	ACC	PPV	NPV	% of Nodules	SEN	SPC	ACC	PPV	NPV	% of Nodules
		UC						SM					
3A-T	3	77.8	83.9	81.7	72.4	87.4	37.8	61.8	100.0	70.5	100.0	43.5	47.7
K-T	5	59.3	93.3	81.3	82.8	80.8	25.2	52.9	100.0	63.6	100.0	38.5	40.9
EU-T	5	77.8	83.9	81.7	72.4	87.4	37.8	61.8	100.0	70.5	100.0	43.5	47.7
Kw-T	4c	61.7	93.3	82.2	83.3	81.8	26.1	52.9	100.0	63.6	100.0	38.5	40.9
ACR-T	5	56.8	93.3	80.4	82.1	79.9	24.3	38.2	100.0	52.3	100.0	32.2	29.5
ATA-T	5	67.9	85.2	79.1	71.4	83.0	33.5	58.8	100.0	68.2	100.0	41.7	45.5
		**FLUS/AUS**						**SFN/SHT**					
3A-T	3	51.4	81.6	78.4	25.0	93.4	21.9	52.6	69.6	67.7	18.2	92.0	32.9
K-T	5	34.3	92.2	86.0	34.3	92.2	10.6	31.6	85.8	79.6	22.2	90.7	16.2
EU-T	5	51.4	81.6	78.4	25.0	93.4	21.9	52.6	69.6	67.7	18.2	92.0	32.9
Kw-T	4c	34.3	91.8	85.7	33.3	92.2	10.9	36.8	85.1	79.6	24.1	91.3	17.4
ACR-T	5	25.7	91.8	84.8	27.3	91.2	10.0	31.6	85.1	79.0	21.4	90.6	16.8
ATA-T	5	45.7	86.1	81.8	28.1	93.0	17.3	36.8	77.7	73.1	17.5	90.6	23.9

**Table 5 jcm-09-02281-t005:** Data on the number and percentage of detected cancers in the whole examined sample (for the threshold values that gave the maximum AUC).

TIRADS/Guideline Threshold Category		No/% of nodules	No/% of cancers	No/% of Cancers ≥ 1 cm	No/% of PTC	No/% of FTC	No/% of HCT	No/% of MTC	OR 95%CI *
3A-T	3	322/32.2 ^d,e^	175/70.0 ^a,b,c,d^	124/65.6	148/74.7 ^a,b,c^	6/37.5	5/38.5	12/75.0	9.6
									(6.9–13.2)
K-T	5	196/19.6	132/52.8	88/46.6	116/58.6	3/18.8	3/23.1	8/50.0	12.0
									(8.4–17.1)
EU-T	5	322/32.2 ^d,e^	175/70.0 ^a,b,c,d^	124/65.6	148/74.7 ^a,b,c^	6/37.5	5/38.5	12/75.0	9.6
									(6.9–13.2)
Kw-T	4c	203/20.3	137/54.8	91/48.1	121/61.1	3/18.8	3/23.1	8/50.0	12.6
									(8.8–17.9)
ACR-T	5	186/18.6	120/48.0	81/42.9	105/53.0	2/12.5	3/23.1	8/50.0	9.6
									(6.7–13.6)
ATA-T	5	271/27.1	153/61.2	107/56.6	134/67.7	6/37.5	3/23.1	8/50.0	8.4
			(16) #	(14) #	(14) #	(2) #			(6.1–11.7)

*, *p* < 0.0001 in all cases; #, cancers in nodules other than hypoechoic; a, *p* < 0.0001 vs. ACR-T; b, *p* < 0.001 vs. K-T; c, *p* < 0.005 vs. Kw-T; d, *p* < 0.05 vs. ATA-T; e, *p* < 0.0001 vs. Kw-T, K-T, ACR-T.

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
