# Peer review of "Diagnostic Value of Six Thyroid Imaging Reporting and Data Systems (TIRADS) in Cytologically Equivocal Thyroid Nodules"

_jcm, 2020, doi:10.3390/jcm9072281_

Round 1

Reviewer 1 Report

The aim of this study was to evaluate the diagnostic efficacy of different TIRADS systems, ACR-TIRADS, EU-TIRADS, Korean-TIRADS, Kwak-TIRADS, AACE/ACE-AME-TIRADS and ATA in nodules with determined and indeterminate cytological diagnosis in subjects exposed to iodine deficiency.

The paper is of interest although some criticism could be raised:

  • In the methods section, the design of the study is not clear. In particular, it is not clear why the authors decided to reach 1000 nodules adding 460 nodules with unequivocal cytologic result. If this choice was due to the power of the study calculation, the methods section should include the reasons followed by the authors to reach this conclusion. Otherwise, if the design of the study resembles a case-control one, a precise strategy to collect controls (for example, with a 1:1 ratio) should be followed and specific matching criteria should be used. Moreover, clear inclusion and exclusion criteria should be reported.
  • Were ultrasound features assigned prospectively at the time of the sonograms or were images reviewed retrospectively for this analysis?
  • Did one person assign all the ultrasound features for TIRADS score calculation? If no, did you consider the inter-observer variability?
  • In Table 1, the p value for each comparison between every category of patients should be reported even in the non-significant cases. Furthermore, the number of equivocal nodules reported in table 1 is 540, while in the text the number reported in 560. Please check
  • To aid the reader, the abbreviations should be spelled out completely at first occurrence and in the abstract
  • Line 330: please check per percentage of cancer
  • In the present form, the paper is too long. I suggest to revise the manuscript and the tables, focusing on the main results
  • English language should be revised

Reviewer 2 Report

General comments:

This paper investigated the usefulness of selected thyroid sonographic risk-stratification systems in the diagnostics of nodules with indeterminate (equivocal) cytology or unequivocal cytology in a population with history of iodine deficiency. The study results suggest that the efficacy of TIRADS depends on the incidence of PTC and all TIRADSs facilitate the selection of FLUS/AUS nodules for the surgical treatment but these systems are not efficient in the management of SFN/SHT nodules.

Major strength

  • Large sample size of indeterminate cytology nodules.
  • Well-organized data analysis and comprehensive review of the relevant literatures.
  • Comparison of the US risk stratification systems of five society guidelines

Major weakness

  • Concern of inaccurate classification of nodules according to each risk stratification system.
  • Unclear clinical implication for management of AUS/FLUS nodules based on this study.

This study is interesting and well-written. However, there are several major methodological issues to be addressed, and several comments and questions are pointed out below, which hopefully could enhance the strength of this study.

Major comments:

  1. Study population

The inclusion and exclusion criteria needs to be clarified as a flow chart in the study cohort of the consecutive data. The reference standard for final diagnosis was the histopathological diagnosis by surgery in this study. Therefore, many nodules which underwent FNA would have been excluded. Authors incorporated 460 nodules with definite FNA diagnosis (category II and VI) to compare the performance of TIRADS with nodules with indeterminate FNA results (category III, IV, VI). Although this provides useful information, it is unclear how authors selected 406 patients (consecutive?) and the prevalence of malignancy of these patients was high (162/460, 35.2%), which would be higher than the actual malignancy risk because surgical diagnosis was used for the reference standard of final diagnosis. The pretest prevalence of malignancy influence on the calculated malignancy risk of each TIRADS (table 3).

  1. Classification of nodules by each risk stratification system.

The US risk stratifications have different definitions of US lexicons, which may result in the different classification of nodules even though they have similar US features. It is uncertain how authors solved this issue. For example, “marked hypoechogenicity” was defined ‘as hypoechogenicity similar to the echogenicity of the strap muscles’ (Table 2). However, the ACR TIRADS and the EU-TIRADS clearly defined “marked hypoechogenicity” as lower echogenicity than the strap muscle. Another important one is the definition of the echogenicity of nodules with mixed echogenicity. Majority of the TIRADS such as ACR TIRADS, K-TIRADS use the predominant echogenicity in nodules with mixed echogenicity, meanwhile, the nodules are categorized as mildly hypoechoic nodules (intermediate risk) if there is  presence of any hypoechoic tissue in the nodule. Why did authors assess the macrocalcification without microcalcification instead of macrocalcifiation separately (regardless of co-existing microcalcification)? The ACR TIRADS adds 1 point for macrocalcification regardless of co-existing microcalcification.

  1. Role of the TIRADS in the management of FNA indeterminate nodules

Authors clearly shows the limited role of the current TIRADS in nodules with FN/SFN FNA results. Although this study verify that the TIRADS could stratify the risk of AUS/FLUS nodules, the malignancy risk of AUS/FLUS nodules with high TIRADS score is around 20-30%, which is relatively lower compared to previous studies. If all those nodules undergo surgery, the unnecessary diagnostic surgery rate is very high (around 70%). It would be reasonable to consider other options including repeat biopsy as recommend by the Bethesda system before surgery.

Minor comments:

  1. The US risk stratification system of the ATA and AACE/ACE/AME guideline is not usually considered as the TIRADS even though the meaning is the same. Those societies currently do not use the terminology of “TIRADS” for their systems and you may consider using other words such as the ATA system or the ATA guideline.
  2. The risk stratification systems of the five societies are widely used and recommended as clinical guidelines by each society, which has somewhat different clinical significance from other TIRADS proposed by many researchers including the Kwak-TIRADS which is rarely used for clinical practice in Korea.
  3. The K-TIRADS was developed based on a multicenter retrospective study [Thyroid Imaging Reporting and Data System Risk Stratification of Thyroid Nodules: Categorization Based on Solidity and Echogenicity. Thyroid. 2016;26(4):562-572], not based on a study by Seo et al., and validated by a prospective multicenter study [A Multicenter Prospective Validation Study for the Korean Thyroid Imaging Reporting and Data System in Patients with Thyroid Nodules. Korean J Radiol. 2016;17(5):811-821], and was adopted as a clinical guideline by the Korean Society of Thyroid Radiology (KSThR) and the Korean Thyroid Association (KTA).
  4. Authors prospectively analyzed US features prior to biopsy. The number of interpreters and experience levels should be clarified.
  5. There are so many abbreviations, which may be confusing to readers. Please minimize the use of abbreviations in the main text. For example, “UMRFs” (the ultrasound malignancy risk features) is not necessary, and the definition is unclear and confusing because the risk stratification systems use different US features and definitions of high risk (or high suspicion) categories.
  6. Table 2. The terminologies of sonographic features need to be clarified. For example, you may consider “taller-than-wide shape” instead of “suspicious shape”. If “suspicious shape” is used, the definition should be described as a foot note. Also, please consider the use of “isolated macrocalcification” instead of “isolated rim calcifications” if this indicates an entirely calcified nodule with complete posterior acoustic shadowing in which no soft tissue component can be identified due to dense shadowing on US [CT features of thyroid nodules with isolated macrocalcifications detected by ultrasonography. Ultrasonography. 2020;39(2):130-136.]. How did authors classify the isolated macrocalcificatons in each TIRADS? The isolated macrocalcification is classified by the K-TIRADS 4 and ACR TR4, but unclassified in other systems [Similarities and Differences Between Thyroid Imaging Reporting and Data Systems. AJR Am J Roentgenol. 2019;213(2):W76-W84.]

Round 2

Reviewer 1 Report

The manuscript was revised along the lines indicated.

Author Response

There is no point to be addressed.

Reviewer 2 Report

I believe that most of raised issues have been successfully addressed, and the revised manuscript has been significantly improved.  

Minor comment:

The various definitions of US descriptors used in various risk stratification systems could make a comparative research difficult, which might be inevitable in a way. There is actually more minor controversy on this issue. For example, the definition of solid composition could be controversial. Although it is acceptable for this study, the definition of ‘pure solid or almost solid composition’ is used for ‘solid nodule’ in the ACR TIRADS and the K-TIRADS, which is different from the definition (cystic component less than 10%) of the EU-TIRADS. Also, the definitions of US lexicons and descriptors are not clearly described in the ATA and AACE/ACE/AME guidelines. Because the ACR TIRADS 2 is identical to the US pattern of ‘partially cystic isoechoic or hyperechoic nodule without other suspicious feature’, more cases of FTC might have been assigned to the ACR TIRAD 2 if stricter definition of solid composition was used in the present study.

Author Response

We do agree with the Reviewer’s opinion that definitions of ultrasound features used in the risk stratification systems are not precise enough. That poses some problems in their practical implementation and is especially apparent when a comparative analysis of these systems is performed. One of these imprecisely defined features is the solid composition. That is why we clearly defined our understanding of that feature as ”solid echostructure: >90% solid”. We did so in spite of the fact that ACR-TIRADS authors indicated in their recommendations: ”Distinguishing solid nodules from mixed cystic and solid nodules may be difficult in practice, as they represent a continuum. Unlike with spongiform nodules, ACR TI-RADS does not require that the observer estimate the percentage of a nodule that is solid, as this determination is often highly subjective and is less important than the characteristics of the solid component.” We do agree with that statement. In clinical practice it is hardly possible to judge whether the cystic component is 5, 9 or 11%. Actually, if we had used a more strict version of solid nodule assessment then nodules with the cystic component in the range of 1-9% would have been treated as ”mixed cystic and solid” and received 1 instead of 2 points. And if there were no other features those nodules would have been categorized as ACR TIRADS 2 and not ACR TIRADS 3. Consequently, sensitivity of nodule qualification for FNA could have been decreased, because ACR TIRADS 2 category is not an indication for FNA. In our study three 3 FTC were classified into ACR TIRADS 3 category, but only one case was of the pattern indicated by the Reviewer. We have reevaluated this nodule and confirmed with the images saved in DICOM format that the nodule was completely solid.